# Knowledge is a Region in Weight Space for Finetuned Language Models

**Almog Gueta** *
Technion - IIT
almoggu@gmail.com

**Elad Venezian**
IBM Research
eladv@il.ibm.com

**Colin Raffel**
UNC Chapel Hill
craffel@gmail.com

**Noam Slonim**
IBM Research
noams@il.ibm.com

**Yoav Katz**
IBM Research
katz@il.ibm.com

**Leshem Choshen**
IBM Research
leshem.choshen@il.ibm.com

## Abstract

Research on neural networks has focused on understanding a single model trained on a single dataset. However, relatively little is known about the relationships between different models, particularly those trained or tested on different datasets. We address this by studying how the weight space and the underlying loss landscape of different models are interconnected.

Specifically, we demonstrate that finetuned models that were optimized for high performance, reside in well-defined regions in weight space, and vice versa – that any model that resides anywhere in those regions also exhibits high performance. Notably, we show that language models that have been finetuned on the same dataset form a tight cluster in the weight space, while models finetuned on different datasets from the same underlying task form a looser cluster. Moreover, traversing around the region between the models leads to new models that perform comparably or even better than models obtained via finetuning, even on tasks that the original models were not finetuned on.

Our findings provide insight into the relationships between models, demonstrating that a model positioned between two similar models can acquire the knowledge of both. We leverage this and design a method for selecting a better model for efficient finetuning. Specifically, we show that starting from the center of the region is as effective, if not more, than using the pretrained model in 11 out of 12 datasets, resulting in an average accuracy improvement of 3.06.

## 1 Introduction

Models that share the same architecture but differ in their weights can have dramatically different capabilities. As an example, finetuned variants of a pretrained model all share an architecture, yet they are specialized for different tasks. This study

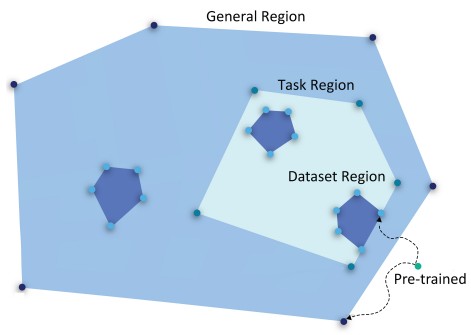

Figure 1: A schematic view of the weight space. Fine-tuning ends up in a *region* determined by the dataset (deep blue) which resides in the task (light blue) and language tasks regions (outer blue). Any combination of finetuned weights is found within the region. Each region is characterized by a low loss on the corresponding: dataset, task datasets, or diverse linguistic datasets. Generally, loss is lower inside the region than outside or in its boundaries.

explores the relationship between the weights of different finetuned models and the capabilities they exhibit. We analyze the *weight space*, where each model is represented by a weight vector $\theta \in \mathbb{R}^n$. For simplicity, we refer to both a point in weight space and the neural network itself as a "model".

We find that distance characterizes models' knowledge and similarity. Particularly, after finetuning a pretrained model on similar datasets, the resulting models are close to each other in weight space (§2.3). Throughout the paper, we consider 3 granularities (§3.1), showing that (i) models finetuned on the same **data** are closer to each other than to other models; (ii) models finetuned on the same **task** also cluster together; and (iii) models finetuned on **general** language tasks are not spread arbitrarily around the pretrained model, but fall in a constrained region in space.

We find that different finetuning runs on the same data tend to converge on similar points in weight space rather than dispersed points. Loosely, those points embed the necessary knowledge to perform

---

*Research done during internship in IBM Research.

the task. This leads to the hypothesis that other points in the proximity of finetuned models might also perform the task well. Notably, such points in weight space might not necessarily be reached via finetuning, but rather via spatial transformations. Indeed, we replicate the finding (Entezari et al., 2021, c.f. §8) that models finetuned on the same dataset are linearly connected, i.e., points on the line between the two models attain similar or even lower loss (§5.1). We expand this finding to the convex hull between the finetuned models (§5.2), suggesting that knowledge is **shared** across the **region** in space. That is, finetuned models define a connected basin of low loss, and every point within it performs well. To show this, we test models sampled from the region and find they even outperform the models achieved by finetuning. Moreover, we replicate the findings in all the aforementioned granularities: regions per dataset, task, and in general. For each, we observe a **low loss across datasets**, beyond the loss the individual models optimized. Furthermore, we show in §6 that these regions are relatively tight, in the sense that extrapolating (rather than interpolating) can quickly produce a poorly performing model.

Our empirical findings have intriguing implications, suggesting, for example, that the best models may not lie at the edges of the region, but rather closer to its center, while finetuning often yields models at the edge of the region. Motivated by these findings, we demonstrate in §7 that a model created by averaging the weights of finetuned models from the same region outperforms the pretrained model on a variety of tasks after subsequent finetuning, even on tasks that the original finetuned models were not trained on.

Overall, our work contributes to the growing body of knowledge about the loss landscape, finding connectivity in a whole bounded region rather than mere linear connectivity, finding connectivity between models not trained on the same task, and finding connectivity in generalization, evaluating models on multiple losses. We also provide initial context to empirical findings about fusing models. We discuss the relations to previous works in §8.

## 2 Experimental Setup

We conduct two main types of experiments. In one we train models with different characteristics (e.g., dataset or task, see §3.1) and examine their representation in weight space using clustering. In the second experiment type, we compare losses of one group of models to another. Below, we describe the datasets (§2.1), settings (§2.2), and granularity levels of comparison between models (§3.1).

### 2.1 Datasets

We finetune and evaluate models on 36 datasets. Those datasets can be categorized into a few families: natural language inference (*NLI*), *Sentiment* analysis and *Topic* classification tasks, *Twitter* domain, and a collection of *general* datasets that covers a wide range of capabilities. We chose classification datasets for reliable evaluation. The details of each dataset family are found in App. A. We mostly rely on the MNLI (Williams et al., 2018b) dataset, the NLI family, and the General group, as case studies, and elaborate on them below:

**General** dataset family contains 12 text classification datasets from GLUE (Wang et al., 2018) and SuperGLUE (Wang et al., 2019), excluding test-only (AX-b (Wang et al., 2019), AX-g (Poliak et al., 2018)) and regression (STS-B (Cer et al., 2017)) datasets. We further exclude WSC (Levesque et al., 2012) and CoPA (Roemmele et al., 2011) which are small and therefore produce unstable results (e.g., finetuning results were sometimes lower than pretrained model results). The datasets consist of a wide range of classification tasks, from sentiment analysis to linguistic acceptability to NLI.

**NLI** family is composed of 6 natural language inference (NLI) datasets: MNLI (Williams et al., 2018a), QNLI Rajpurkar et al. 2016, RTE (Dagan et al., 2005; Bar-Haim et al., 2006; Giampiccolo et al., 2007; Bentivogli et al., 2009), WNLI (Levesque et al., 2011), ESNLI (Camburu et al., 2018), and adversarial NLI (Nie et al., 2020).

### 2.2 Training Approaches

We experiment with RoBERTa-base (Liu et al., 2019) as our base pretrained model, except in App. B where we analyze different pretrained models. For finetuning, we follow the standard hyperparameters (Liu et al., 2019), with a larger batch size of 256 and a learning rate of $5e-5$. Most experiments analyze 5 different seeds, and the same-dataset clustering 20 seeds (§3.1). Those seeds control randomly initialized weights in the classification head as well as data shuffling.

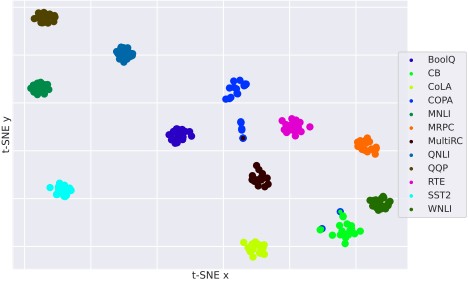
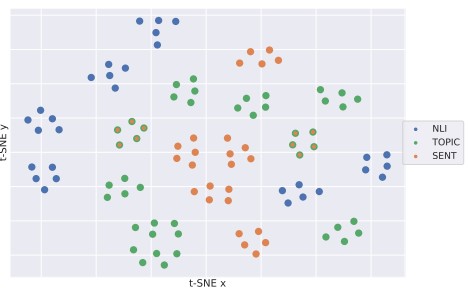

| (a) Clustering models by dataset. | (b) Clustering models by dataset family. |

Figure 2: Clusters of finetuned models on different datasets or tasks, projected by t-SNE. We find that both datasets and dataset families correspond to regions in space. In each figure, each model is represented as a dot, where the inner color is the color of the dataset/task the model was finetuned with and the outer color is the color of the most common dataset/task in the cluster (representing the cluster label). Datasets/tasks names are shown in legends.

## 2.3 Clustering Approach

In the clustering experiments, we qualitatively explore whether models trained on similar data end up close together in weight space. We experimented with various distance metrics and clustering algorithms. While many metrics worked well, we found that subtracting the pretrained weight values from the finetuned values (referred to as "task vectors" by Ilharco et al. (2022)) and measuring distance via cosine similarity was conceptually simple, cheap to compute, and provided qualitatively reasonable results compared to more sophisticated methods (Kornblith et al., 2019; Toledo et al., 2022). We also tested Euclidean distance but it did not produce clear clusters. This is likely caused by the weights' norm growth during training (Merrill et al., 2020) that is unrelated to the data at hand (§C). This can also explain questions that were previously left open (Qin et al., 2022). As a clustering algorithm, we use Spectral Clustering with as many clusters as datasets or dataset families (Pedregosa et al., 2011). For visualization, we project the 120M dimensional weight vectors into 2 dimensions using t-SNE (Van der Maaten & Hinton, 2008).

## 3 Methodology: Comparing Models

In this work, we compare models that share an architecture, but were trained on different data. To do so, we investigate the space of weights $\omega \in \mathcal{R}^d$ where each model has a weight vector and each point in space represents a model. We adopt the typical perspective that the model $f_\theta$ consists of a representation encoder $f_\omega$ followed by a task-specific classifier $f_\phi$, i.e. $f_\theta = f_\phi \circ f_\omega := f_{\phi,\omega}$ (Choshen et al., 2022a; Ram'e et al., 2022).

Ideally, we would compare finetuned models by

their loss. Unfortunately, the loss is often incomparable across datasets or tasks. Hence, we compare by preserving each encoder, and fitting a classification head to each model for each target dataset.

Specifically, to calculate the loss of a model we perform the following: First, we remove any existing masked language modeling layers or classification heads and replace them with a new randomly initialized classification head. This leaves the rest of the weights i.e., the encoder $f_\omega$, fixed. We then perform *linear probing*, i.e., we train only the new classification head on a desired target data $x_{train}$ and its labels $y_{train}$. Lastly, we pass the test data $x_{test}$ through the model (including the classifier $f_\phi$ on top of it) and report the loss with respect to the labels $y_{test}$. Formally, for the model $f_{\phi,\omega}$ and loss function $l$, we report the generalized loss $l_g(\omega) = l(f_{\phi,\omega}(x_{test}), y_{test})$ where $f_\phi = \arg\min_\phi l(f_{\phi,\omega}(x_{train}), y_{train})$. This approach has a desirable trait: When considering the task on which the model was originally finetuned, our loss $l_g$ is equal to the original finetuning loss $l$. Furthermore, since fitting a linear classifier given a fixed representation is a convex optimization problem, we observe similar results across runs.

The *generalized loss $l_g$* enables comparing models finetuned on different datasets. It is hence undesirable to test only on one of the datasets. We thus consider a loss on a dataset, but also the average loss on a family of datasets. For example, the average loss across all entailment datasets rather than the loss on a particular dataset.

### 3.1 Levels of Granularity

To study the relationship between weights of similarly trained models, we experiment with 3 levels

of granularity for dataset similarity. At each level, we analyze models finetuned on source datasets sharing some traits. In each level's setting, we define an *interior* group (hereafter *In*) of datasets that share a trait as well as an *exterior* group (hereafter *Ex*) of models not sharing the trait. By default, we report on each group the average loss over all source datasets used for finetuning *In* models.

**Same-Dataset.** In the most specific case, models are similar if they were finetuned on the same dataset. Interior models are finetuned on MNLI (Williams et al., 2018a) and *Ex* on the rest of the General datasets. We report the loss over MNLI.

**Same-Task.** At this broader granularity, we consider the group of models trained on the same task. In that case, *In* contains models finetuned on NLI datasets and *Ex* contains models finetuned on all other datasets. We report loss over all NLI datasets, except for ANLI which is not intended for such test purposes. ANLI is made with adversarial examples that cause misclassifications for NLI-trained models. In initial trials, it showed similar trends, but we omit it from the test for good practice.

**General.** In the most general case, we consider any model finetuned on any of the General datasets as *In*. This leaves little to consider as exterior, so we construct *Ex* by perturbing the pretrained model's weights in a random direction. We apply a perturbation whose norm is equal to the average task vector norm of *In* models. Since there is no clear prior to sampling a random direction in space, we aim for a prior that prefers points in the weight space that represent "reasonable" networks. We use Xavier initialization (Glorot & Bengio, 2010) to define such a prior. The prior is an i.i.d. Gaussian distribution over each weight with zero mean and where variance depends on the layer characteristics. This choice reduces the probability of sampling networks with exploding or vanishing outputs, which would stand as a weak baseline.

## 4  Analysis in Weight Space

We start our analysis by showing that the models trained on **similar data** fall into the **same region** in weight space - i.e., they are clustered together. We leave the inverse claim (i.e. showing that models within the cluster obtain a lower loss than the models outside the cluster) to §5.1 and §5.2.

Specifically, we find (see Fig. 2) that finetuning on similar data results in closer weight space mod-

els compared to models that have been trained on different datasets or tasks. Notably, despite the fact that neural networks implement highly non-linear functions, finetuning similarity is expressed in the Euclidean space of their weights. Moreover, we show in App. §C that the direction in space is determined by the type of training data and not by its amount. In App. B, we show that this proximity is contingent on starting from the same base model.

**Similarity Per Dataset.** In the simplest case, for each dataset in the General group, we finetune models with 20 random seeds and cluster the resulting 280 models into 12 clusters. As seen in Fig. 2(a), for the most part, models finetuned on the same dataset are clustered together. Accordingly, the overall clustering accuracy is 98%, with all but 3 clusters perfectly matched.

**Similarity Per Task.** In this experiment, we show that models finetuned on datasets from the same task are also close in weight space (we discuss same-domain proximity in App. D). As explained in §2.1 we have dataset families for 3 tasks: NLI, Topic, and Sentiment. For each dataset in each family, We finetuned models with 5 random seeds. Then, we cluster all models into 3 clusters. As seen in Fig. 2(b), models that were finetuned on the same task family are closer to each other and are clustered together (clustering accuracy of 90%). We report the $F_1$ Score per group in App. D.

**Similarity in General.** Unlike datasets or tasks, we can not create multiple distinct general groups and can not expect multiple clusters to occur. Therefore, we do not present clustering for this granularity level. However, we can still infer that this general region does not encompass the whole space around the pretrained model, and has a superior loss in general (see §5.2).

### 4.1  Cause: Data Type, not Size

Supposedly, a confounding factor may explain the above results, wherein the finetuned model moves more with more data. To test this, we finetune models on sub-samples with different sample sizes (200, 400, 800, 1.6K, 3K). For consistency, we take only the 9 datasets from General family that contain at least 3K training samples. We then cluster the finetuned models into $k$ clusters, with $k$ the number of datasets or the number of dataset sizes.

The resulting clusters (App. C) are clustered by data type, not by the amount of data, similar to

Fig. 2. Choosing $k$ to be the number of data-sizes does not cluster by data size either. We conclude that the observed similarity comes from the nature of the data, and not from the size of a given dataset.

## 5 Loss in the Region between Models

In §4, we claim that models trained on similar data converge near each other, but is this area to which they converge meaningful? In this section, we show that models falling in the entire region around these clusters correspond to performant models.

The models we analyzed so far were the outcome of a gradient-based optimization process searching for the minimum loss. The locality we observed in weight space indicates that the points found through this procedure are concentrated in relatively small regions. We hypothesize that a whole region of low losses (corresponding to performant models) exists between the separate points found during finetuning. For example, the "NLI region" contains MNLI, SNLI and QNLI models but also other points that reflect models that might not have been found through gradient-based optimization on a specific dataset but exhibit the general abilities needed to perform natural language inference.

We test this hypothesis by interpolating pairs of similarly trained models and show in § 5.1 that the points between the models perform comparably to or even better than the original finetuned models. This suggests that indeed there are regions in weight space where all points encode the knowledge or behaviour required for a particular task. We expand this claim in §5.2 and show that the whole region that lies between these models (their convex hull) corresponds to models that perform well.

### 5.1 Interpolation: Lines Between Model Pairs

In this experiment, we consider the points in weight space between pairs of finetuned models. Given a pair of models, we shift from one model to the other by linearly interpolating between their weights, i.e., we take the model's weights $\omega_1, \omega_2 \in \mathcal{R}^d$, and consider weighted sums of their weights: $\omega_1 * \alpha + \omega_2 * (1 - \alpha)$. where $\alpha \in [0,1]$. We then evaluate each interpolated model both on the datasets the original models were finetuned on, and on additional datasets unseen by the models. We interpolate pairs of different models finetuned on the same dataset, or on two different datasets. We report the average losses produced by repeating the experiment with finetuning using different seeds.

Results ( Fig. 3) show that interpolated models perform comparably or even better than the models they are created from. We present further results testing the groups on different losses in App. §E and find performance is often best somewhere in the interpolation between the two models. We now elaborate on each granularity level separately.

**Interpolation Per Dataset.** We interpolate 5 finetuned models on the MNLI dataset (resulting in a total of 10 pairs) and evaluate on MNLI. We report an analogous experiment with SST2 in App. §E. Figure 3(a) shows that the interpolated models perform well on average and even outperform the original models they are created from. Similar results were found in other settings (e.g.; Wortsman et al., 2022b) and we discuss those works in §8.

**Interpolation Per Task.** We interpolate 5 models finetuned on MNLI with 5 models finetuned on ESNLI, both from the NLI task, resulting in 25 pairs, and evaluate on all NLI test datasets. We replicate the results of the previous experiment and find the interpolated models are performant on all targets on average, as can be seen in Fig. 3(b).

**Interpolation In General.** We interpolate 5 models finetuned on MNLI with 5 models finetuned on SST2, both from the General family, resulting in 25 pairs and evaluate on all General datasets as targets. Fig. 3(c) shows improved performance in this extended group and better performance in the interpolated models than in the finetuned ones.

### 5.2 Comparison between Region losses

Thus far, we showed that models on the line between model pairs perform well. We now extend the analysis to show that models in the whole region between similar models perform well. However, visualizing or searching a whole multidimensional region (the convex hull) is not feasible. Instead, we sample models in the region and show they outperform their external counterparts.

Let *In* be a group of models and *In'* be the convex hull between all the models in *In*, making each model in *In'* a weighted average of models in *In*: $\sum_{i=0}^{|In|} \alpha_i \cdot \omega_i$ where $\sum_{i=0}^{|In|} \alpha_i = 1$ and $\omega_i \in$ In. Practically, as *In'* is infinite, we estimate it by sampling $|In|$ models uniformly from the region they convey.

We note that weighted averaging in this manner was shown to be practical and work well in many scenarios, either in efficient finetuning (§7 Yadav

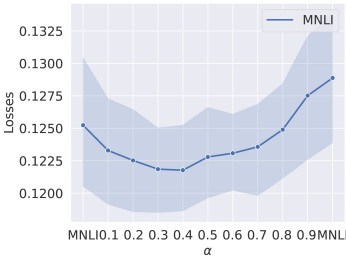

(a) Interpolation per dataset.

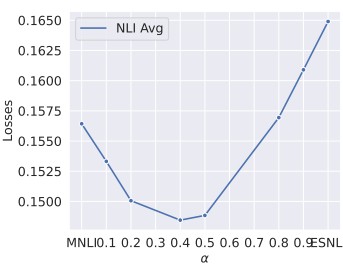

(b) Interpolation per task.

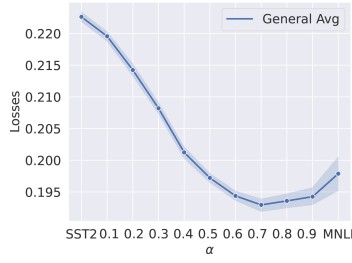

(c) Interpolation in General.

Figure 3: Losses of linearly interpolated models created between pairs of similar models. The best loss often lies between models. In each figure, the solid line is the losses' average during interpolations for different $\alpha$ values, the edges of the lines represent the average loss pure finetuned models we interpolate, the Y axis is the average loss value, and the X axis is the position determined by $\alpha$. The shade is the standard deviation of the losses' average.

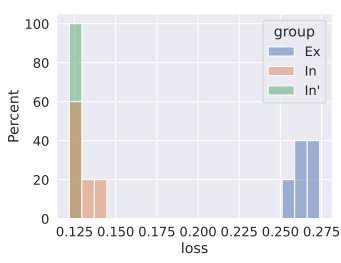

(a) Losses in the dataset region

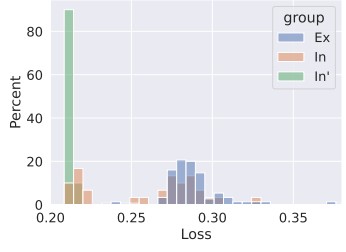

(b) Losses in the task region

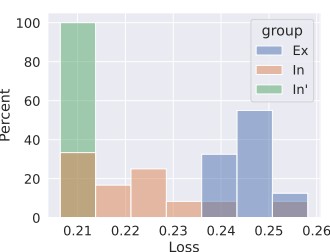

(c) Losses in the general region

Figure 4: Loss distributions of 3 groups: *In* (similarly finetuned models), *In'* (models between models in In), and *Ex* (baseline models). Fig. 4(a) shows 5 models from MNLI region tested on the MNLI loss. Fig. 4(b) shows models from NLI region tested on NLI losses. Fig. 4(c) shows models from the General region tested on the General losses.

et al., 2023) or in full finetuning (Choshen et al., 2022b; Matena & Raffel, 2021, c.f. §8).

We define a metric to compare two groups of models. Given *In* models group and the exterior models group *Ex*, we calculate $PB$ as the probability that an *In* model outperforms an *Ex* one:

$$PB = \mathop{\mathbb{E}}_{i \in \text{In}, j \in \text{Ex}} \mathbb{1}\{l_g(\omega_i) \leq l_g(\omega_j)\}.$$

$PB$ can also be applied to *In'* and *Ex*.

As a loss function, we take the average loss over the source datasets used to create the *In* models.

Testing models from *In* and *In'* groups, we find they indeed outperform *Ex* models on the tasks the *In* models were trained on. We find this is true in all granularity levels – models in the dataset region are better than other models, and more broadly *any* finetuned model is better than models that have been randomly shifted by the same distance from the pretrained model. Moreover, we again find (as in §5.1) that *In'* is even better than the *In*. In addition to the bottom-line metric PB, we depict the loss distributions across those models in Fig. 4.

**Loss Per Dataset.** We test the performance of models between models finetuned on a dataset. We consider the case where *In* is the group of finetuned models on MNLI and *Ex* is the group of finetuned models on General datasets. Both groups are evaluated on the MNLI dataset. We find PB is 100% for *In*, meaning that all MNLI models outperform on MNLI than all the rest of the models. More surprising is that the same is true for *In'*, PB of 100% – all the models between MNLI models are better than *Ex*. In fact, in 88% of the times *In'* models are also better than *In* – i.e. models finetuned on MNLI!

**Loss Per Task.** We compare models from a task region with models from other regions. Here, *In* are the models finetuned on NLI task and *Ex* on the rest of the datasets described in §2.1. Both groups are evaluated on all NLI test datasets. NLI *In* group models are better in $PB = 75.3\%$ of the cases, and the *In'* models in 100%. *In'* is also better than *In* with $PB = 96.7\%$.

**Loss In General.** We define *In* to be finetuned models on General datasets and *Ex* to be random models as defined in §3.1. Both are evaluated on

the General datasets. We find again that *In* is better than *Ex* ($PB = 89.8\%$) but worse than *In'* ($PB = 90\%$) which is also better than *Ex* ($PB = 100\%$).

To conclude, we consistently see that the region between finetuned models not only provide models that are better than the baseline but also provides models that are better than the finetuned models defining the edges of region.

# 6 Region Edges

Above, we have shown that there are spacial regions that specify learnt generalizations. We now look for the boundaries of those regions, where loss is no longer similarly low. To do that we traverse in the opposite way to the interpolation. We also test the edges going from the center of the region to other directions in App. F.

## 6.1 Extrapolation: Lines Between Models

In Section 5.1, we took pairs of models and found that the linear path between them passes through a region of low loss. We now continue on this path and check how far in the opposite directions (i.e. away from the model being interpolated to) do we need to move in order for the loss to rise. We reproduce the interpolations settings of §5.1, but apply linear *extrapolation*, i.e., test $\alpha$ values out of range [0,1]. We make 10 steps in logarithmic advances from 1 to 32 and similarly from 0 to -31.

Figure 5 depicts the results for the Same-Dataset granularity level. We provide more detailed results in App. G. We find that for all granularity levels extrapolation rapidly reaches bad performance. This implies the converged models are near the edge of the loss basin. We further observe that the region has a relatively flat base and steep cliffs, implying that the regions we find are small basins and not e.g. a subspace. In a sense, we discover a bounded region that characterizes the loss region (of e.g., MNLI dataset) where the models within have a low loss and the models beyond have a high loss.

# 7 Practical Takes

Our work has several practical implications. First, we observed (§5.2) that models inside the region (*In'*) are often superior to the finetuned models defining the region (*In*). Practically, one can average models from the same region and cautiously expect the resulting model to perform better. This model can be used without further finetuning, in

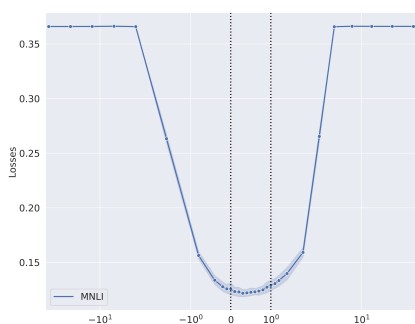

Figure 5: Losses of linearly extrapolated models created from pairs of models finetuned on MNLI. The solid line is the average losses, the vertical dashed lines indicate the average loss of the pure models we extrapolate ($\alpha = 0$ or $\alpha = 1$), and the X axis is the position (meaning the $\alpha$ and $(1 - \alpha)$ values used in the extrapolation). The shade is the standard deviation across runs.

the Same-Dataset region, as has indeed been used in practice (c.f. §8; Wortsman et al., 2022b,a).

We provide another implication of our findings. If indeed models in *In'* share partial information with models from *In*, this aggregated information may be general and useful for other tasks. In practice, there are two common uses for a trained model, either for the immediate classification of unseen examples or as a starting point for further training. We focus on the later use as a low loss directly indicates it could be useful in that setting.

We hypothesize that points in the region could be better for finetuning than finetuning the pretrained model itself. As there are endless possibilities of points in the region with no preference to any specific, we practically pick the centroid of the region, i.e., the average between models in *In*. The centroid point is equally influenced by each model defining the region, and without further information may be stronger than arbitrary points in the region (see App. §E), but also be suboptimal (see §5.1, App. §E).

For subsequent training, we employ parameter-efficient finetuning. Specifically, BitFit (Ben Zaken et al., 2022), one of the most parameter-efficient methods, which has been shown to attain strong performance. Changing only a small subset of the weights reduces the complex effects of training dynamics and eases attributing improvements to the initialization weights. We avoid giving an unfair advantage to our method and for each target dataset choose the centroid of all models excluding ones finetuned on the target dataset itself.

We find (Fig. 6 and App. H) that starting from

the centroid results in a better performing model than starting from a pretrained model, by 4.04% on average. The centroid is better in almost all cases, outperforming the pretrained in 9 cases, matching the results in 2, and underperforming in 1 case.

Efficient finetuning is especially interesting in the scenario of scarce data (App. H). We hence replicate the results in a few-shot scenario limiting the training examples to 1K. The general trend is replicated, only that improvements reach as high as 34% improvement and above 10.66% on average.

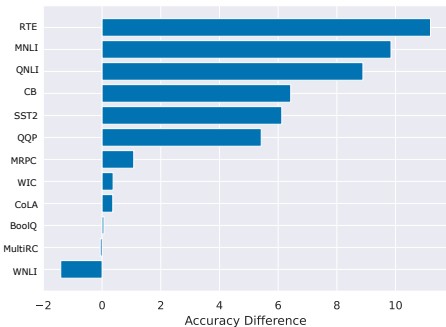

Figure 6: Centroid model gains over the pretrained. Models efficiently finetuned(BitFit) over target datasets.

## 8 Explaining previous results

A great deal of prior work considered the connectivity between models, i.e. whether the path in weight space between two networks has a low loss throughout. Early work demonstrated that models trained on the same dataset have such a path but that the path is not necessarily linear (Garipov et al., 2018; Frankle et al., 2020). This non-linearity was often explained by the fact that networks can represent the same function after their weights are permuted (Ainsworth et al., 2022; Jordan et al., 2022; Chen et al., 1993; Hecht-Nielsen, 1990). Taking into account these symmetries and/or using the same initialization was then shown to produce a linear path of low loss (McMahan et al., 2017; Entezari et al., 2021). Benton et al. (2021) even considered simplexes of low loss, rather than linear paths. In addition, Mirzadeh et al. (2020) showed that multitask learning converges to a point with low loss for both tasks, and in parallel work Qin et al. (2022) showed that the minima are connected for two datasets of the same task. We generalize those notions in the context of finetuned models. Specifically, we confirm that indeed there is a linear path between two models, but further that there is a whole region with low loss through which the

linear path moves. Intriguingly, we have observed that these low-loss regions are unique for each specific dataset or task, whereas Juneja et al. (2022) has reported the existence of multiple basins per each. We also generalize this finding to models that were not trained on the same data and are tested on different data. Qin et al. (2022) is the only work we know to compare models trained on different tasks. However, they report random chance performance in this case. To enable meaningful model comparison, we proposed the generalized loss (§3).

Our results also support and provide some preliminary explanations of recent practical findings. Some works show that starting from a finetuned model helps when finetuning on a different target dataset (Choshen et al., 2022a; Phang et al., 2018), which may be related to the fact that the initial finetuning stage moves the model into the general "language" region (or, even better, the region of space corresponding to the target task). Moreover, a growing literature has shown improvements from averaging two or more finetuned models. Some of those average models trained on the same dataset (Wortsman et al., 2022b,a), which we show picks a model from inside the dataset region. Others show that averages between models can improve models from tasks that they were not trained on (Choshen et al., 2022b; Matena & Raffel, 2021), which agrees with our more general findings. Ilharco et al. (2022) further suggests that some attributes can be added to the model by moving in certain directions in the weight space. In parallel work, Ram'e et al. (2022) considers two finetuning stages before averaging. Lu et al. (2022) and Talman et al. (2023) propose optimization methods featuring Stochastic Weight Averaging (SWA). Our results may indicate that the success of such methods may be partly attributed to its tendency to fall within a region, rather than on its borders. More recent work considers iterative model averaging, where in each iteration multiple models are trained in parallel from the same initial point and then averaged to aggregate their knowledge. Such a procedure has been demonstrated both for self-supervised pretraining (Li et al., 2022) and as a supervised pretraining, similar to a massively multitask learning scenario (Don-Yehiya et al., 2022). Future work could focus on understanding how those processes move through the weight space and whether they move to areas of loss space outside of the region corresponding to a single iteration of averaging finetuned models.

## 9 Conclusion and Discussion

Combining all of our results together conveys a consistent message: There are regions in weight space corresponding to good performance on a dataset, a task, or in general. From §2.3 we can conclude that performant models are centered in certain areas (or more specifically basins) in weight space. We find in §5.1 that these form one basin rather than multiple nearby points falling into multiple basins and, in §5.2, that this basin is a convex region and not simply a line between two points. Finally, the extrapolations in §6 show those areas do not exceed far beyond the finetuned models. Moreover, our results suggest that models found via finetuning typically lie on the boundaries of these regions and are often suboptimal, prompting future work in exploring the limitations of gradient-based training.

## 10 Limitations

We discuss limitations where relevant throughout the work, but also provide this section for general discussion of limitations.

Our work was only evaluated on finetuning a pretrained model, and hence may not hold in general when randomly initializing. They also focused on English classification data.

While our results were very robust when referring to tasks, we did not find many groups of datasets of distinct domains to test on and got mixed results in those aspects. We discuss the results in App. D.

The scope of our experiments is broad in some aspects it is less so in others. While our experiments included thousands of finetuned models, trained on 36 datasets and also evaluated on 36 datasets. We did not replicate it on many pretrained models as well.

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

## A Dataset List

Most datasets could be downloaded from huggingface datasets. We explicitly state the download link when relevant. As we used groups of datasets we report here the full list of datasets they contain.

General: CoLA (Warstadt et al., 2019), SST2 (Socher et al., 2013), MRPC (Dolan & Brockett, 2005), QQP (`data.quora.com/First-Quora-Dataset-Release-Question-Pairs`), MNLI (Williams et al., 2018a), QNLI Rajpurkar et al. 2016, RTE (Dagan et al., 2005; Bar-Haim et al., 2006; Giampiccolo et al., 2007; Bentivogli et al., 2009), WNLI (Levesque et al., 2011) BoolQ (Clark et al., 2019), CB (de Marneffe et al., 2019), CoPA (Roemmele et al., 2011), MULTIRC (Khashabi et al., 2018), WIC (Pilehvar & Camacho-Collados, 2019)

NLI datasets: MNLI (Williams et al., 2018a), QNLI Rajpurkar et al. 2016, RTE (Dagan et al., 2005; Bar-Haim et al., 2006; Giampiccolo et al., 2007; Bentivogli et al., 2009), WNLI (Levesque et al., 2011), ESNLI (Camburu et al., 2018), adversarial NLI (Nie et al., 2020).

Twitter domain datasets (collected by TweetEval (Barbieri et al., 2020)) EmoInt (Mohammad & Bravo-Marquez, 2017), Emoji (Barbieri et al., 2018), Irony (Van Hee et al., 2018), OffenseEval (Zampieri et al., 2019), HatEval (Basile et al., 2019), Sentiment Analysis (Rosenthal et al., 2017)

Sentiment Analysis: Poem Sentiment (Sheng & Uthus, 2020), IMDB (Maas et al., 2011), Rotten Tomatoes (Pang & Lee, 2005), SST 5bins (Socher et al., 2013), SST2 (Socher et al., 2013), Amazon reviews (He & McAuley, 2016) ,Financial Phrasebank (Malo et al., 2014)

Topic Classification: AG news (Zhang et al., 2015), ISEAR (Scherer & Wallbott, 1994), Yahoo answers (Zhang et al., 2015), DBpedia (Zhang et al., 2015), 20 newsgroup (Zhang et al., 2015), TREC in both fine-grained and coarse-grained labels (Li & Roth, 2002)

## B Similarity Per Dataset, when Starting from different Pretrained Models

After seeing in §2.3 the repeated behavior on several granularity levels, we were curious whether we could receive the same behavior on a larger granularity level - models starting from different pretrained RoBERTa models, and finetuned on the same datasets. In this experiment, we employ two pretrained RoBERTa models, the original RoBERTa-base and the re-implementation of RoBERTa-base created by Elazar et al. (2022). We finetune each one on the same datasets, from the General family. Results show that the models get clustered according to the pretrained model they were created from, regardless to the finetuning they went through. This might arise from the low distances moved from the initialization, pretraining changes the model's weights much more than finetuning. Therefore, since we start from different pretrained models, the resulted finetuned models are more similar to the pretrained model they started from.

As the results on both pretrained models are comparable, we deduce that there is not one unique basin or region for each ability, but many. However, around a starting point it seems there are distinct regions within reach.

## C Cause: Data Type, not Size

We provide the clustering and visualize with t-SNE in Fig. 7. We see that the clustering and the data type agree in all but one of the cases.

We provide in Fig. 8 a detailed view of the similarities between each pair of models by dataset and amount of data seen in training. We find that with relatively little data, the direction in space is already determined, i.e., similar datasets go to similar direction even with limited amount of data.

## D Similarity Per Task and Domain

As noted in 2.1, the datasets we use can be separated into specific four dataset families in addition to the general group: NLI, Sentiment analysis, Topic, and Twitter. while the first three are characterized by their task, the last group is characterized by the *domain* of the dataset it contained. As one can see in Fig. 9 and 1 although the clustering shows good separation between task groups, it struggles to separate the Twitter domain group models from the other groups. Separating the space into 4 clusters and labeling them in a

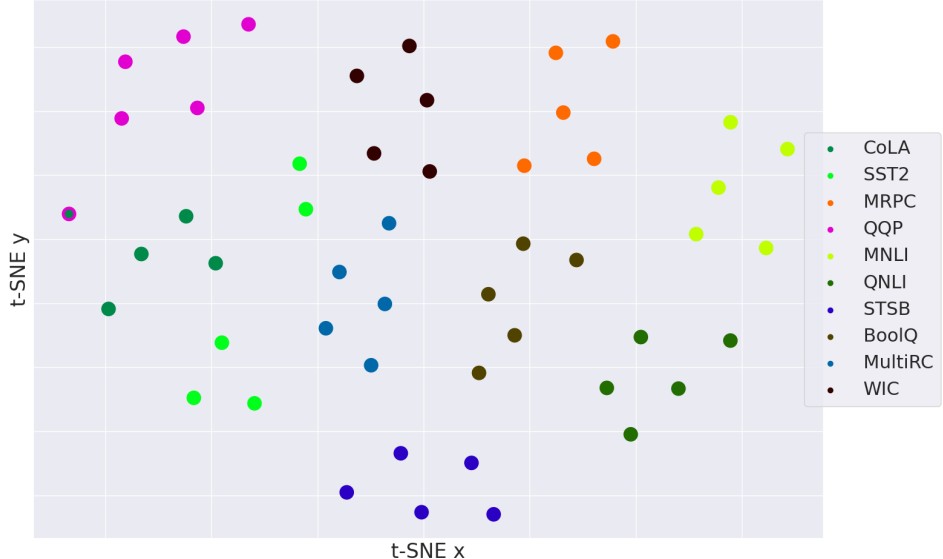

Figure 7: Clusters of finetuned models on different datasets, with increasing train set sizes, projected by t-SNE. Each model is represented as a dot, where inner color is the color of the dataset the model was finetuned with, and outer color is the color of the most common dataset in the cluster (representing the cluster label). Datasets names are shown in legend.

| Experiment/class | Twitter | NLI | Topic | Sentiment | Avg |
|---|---|---|---|---|---|
| F1 Cluster Tasks and Domain | 30 | 100 | 61 | 71 | 65 |
| F1 Cluster Tasks | | 100 | 87 | 83 | 90 |

Table 1: $F_1$ Score - Classification performance by cluster majority. In columns, model group names, in rows the two clustering settings, with and without the domain group (Twitter).

1-to-1 mapping to maximize accuracy, we find 31 f-score on the Twitter cluster and 62,71,1 on the Topic, Sentiment and NLI groups respectively.

A possible explanation may be that the domain feature is orthogonal to the task feature, in the sense that some datasets should be assigned to two groups at the same time (for example TweetEval Sentiment Analysis (Rosenthal et al., 2017) is part of the Twitter domain group, as well as the Sentiment analysis task group). This gives place to two competing hypotheses that we leave unanswered. Either the regions of domains overlap with regions of tasks; or, even if less likely, domains are not separated into regions in space, unlike tasks.

## E  Interpolation Between Models

We provide a more comprehensive interpolation experiment. In it we show the interpolation between pairs of models and report the loss of each of the datasets used to create the pair of models, as well as the average reported in the main paper.

In Fig. 10, one can see not only the interpolation between models in *In*, but interpolation between the centroids. We take the average of all the models in one group from which we interpolate (e.g., all MNLI models) and set it as a centroid. We then repeat it on the other group and interpolate between those centroids instead of interpolating between actual finetuned models. We find that although now we are interpolating between two points that were both not the outcome of traditional ways of optimization, we find comparable and often even lower losses than before. This also motivates the practical experiments reported in §7.

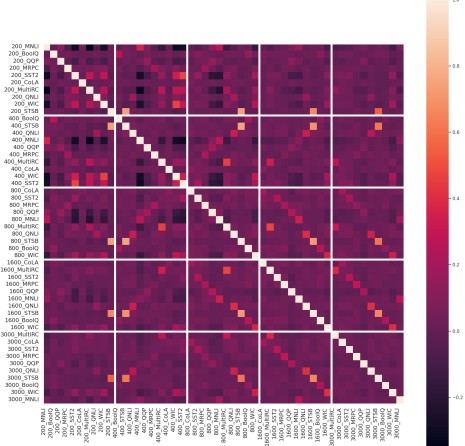

Figure 8: Cosine similarity between models trained on different datasets, with varying data sizes (blocks). The diagonal per block is blurred at the beginning of training, but with still a small amount of data models are highly similar to models trained on similar data. We do not observe similarity between models of similar size.

## F  Loss Region Outside of Models in Other Directions

After seeing that we can reach outside of regions by performing linear-extrapolation, we test the performance of models when we move away to different directions. To test it, we start with several models of the same region, calculate their centroid by averaging their weights, and gradually move away from this centroid according to the same procedure as in section 3.1. We move away from the centroid towards one of two directions: towards the origin of the axis, or towards random directions. We evaluate on the same datasets the *In* models were finetuned on.

Figure 11 shows the results for the first and third granularity levels.

A detailed analysis for each level follows.

**Outside of the Dataset Region.**   We compare the performance of three types of models: finetuned models on MNLI, models moving from the centroid of MNLI models to the origin, and models moving from it to random directions.

Results show that when the distance of the generated models from the centroid is similar to the distance of the finetuned models ($radius \leq 1$), the generated models perform as well as the finetuned models, meaning we are still inside the MNLI region and all models share the knowledge needed for the MNLI target task. It also implies the directions in which finetuned models vary are not special, most changes around the center are equally acceptable.

When the distance increases and we move farther away from the centroid, the performance of the randomly generated models decreases significantly, indicating the end of the region. A surprising finding is that this happens on random directions, but not when going towards the origin. The performance in that case is similar to the performance of the finetuned models, even when the models are farther from the centroid then the finetuned models. While we did not expect this phenomenon or have an explanation to it, we report it as an avenue for future work.

**Outside of the Finetuning Region.**   We compare the performance of three types of models: finetuned models on datasets from the General family, models starting from the centroid of those models towards the origin or towards random directions. Each time, we evaluate all above models on a single dataset from the General family, separating the performance of the model finetuned on the target dataset (called source model), to the rest of finetuned models (called non-source models), resulting in total of four types of models in the comparison, including the two types of generated models starting from the centroid.

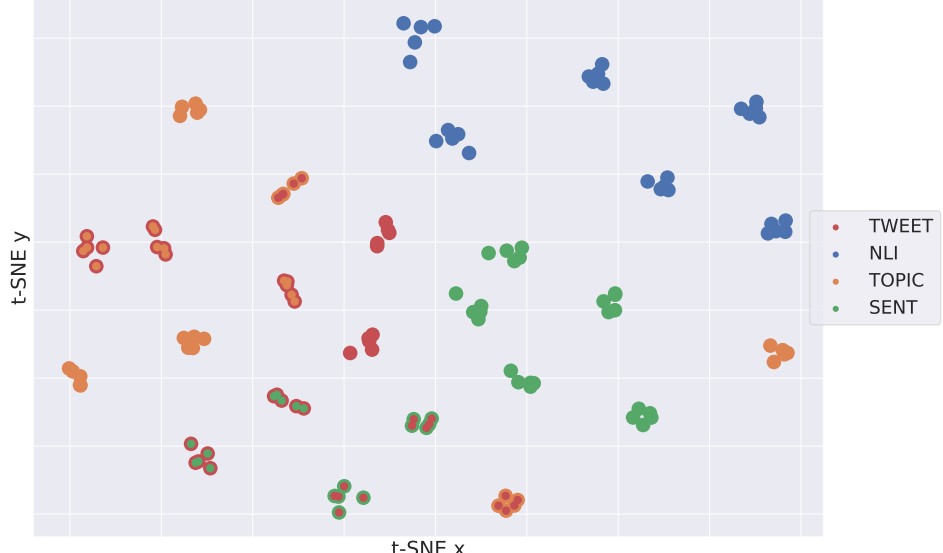

Figure 9: Clusters of finetuned models, trained on datasets groups, distinct by task and domain. The models projected by t-SNE, where each model is represented as a dot, where the inner color is the color of the task/domain the model was finetuned with and the outer color is the color of the most common task/domain in the cluster (representing the cluster label). We find that tasks are can be easily distinguished, while it is hard to separate Twitter domain models.

We average the performance of each type on all target datasets we evaluate on, and show the results in Figure 11(b). We can see that the source model outperforms all other models. For small distances from the centroid, the non-source models underperform the generated models, and for large distances it outperform the generated models going towards random directions. The generated models going towards the origin outperform the two above types of models, for all distances. These results suggest that when staying close enough to the centroid, roaming from the centroid to different directions might be superior to a finetuned model on a different dataset. However, when distancing far from the centroid, finetuned models on other datasets then the target dataset perform better than generated models going towards random directions, since the last are probably outside of the region. Worth noticing, the standard deviation of the last is meaningfully larger than the rest of the models, and of the one of generated models in the Dataset granularity level.

## G   Extrapolation Between Models

Fig. 12 presents the same behaviour for all three granularity levels- extrapolation rapidly reaches bad performances.

We provide a more comprehensive extrapolation experiment showing each time the extrapolation with the loss of each of the datasets used to create the pair of models, and the average reported in the main paper. We find (see Fig. 13(b)) that despite all of our datasets called and framed as natural language inference, WNLI (Levesque et al., 2011) behaves differently and might be considered not strictly a part of the region. This may also explain the long tail in Fig. 4(b).

## H   Efficient Finetuning

We provide in this section the full results of efficiently finetuning. We provide the full results of the regular finetuning in Table 2 and the few-shot setting in Table 3 and Fig. 14.

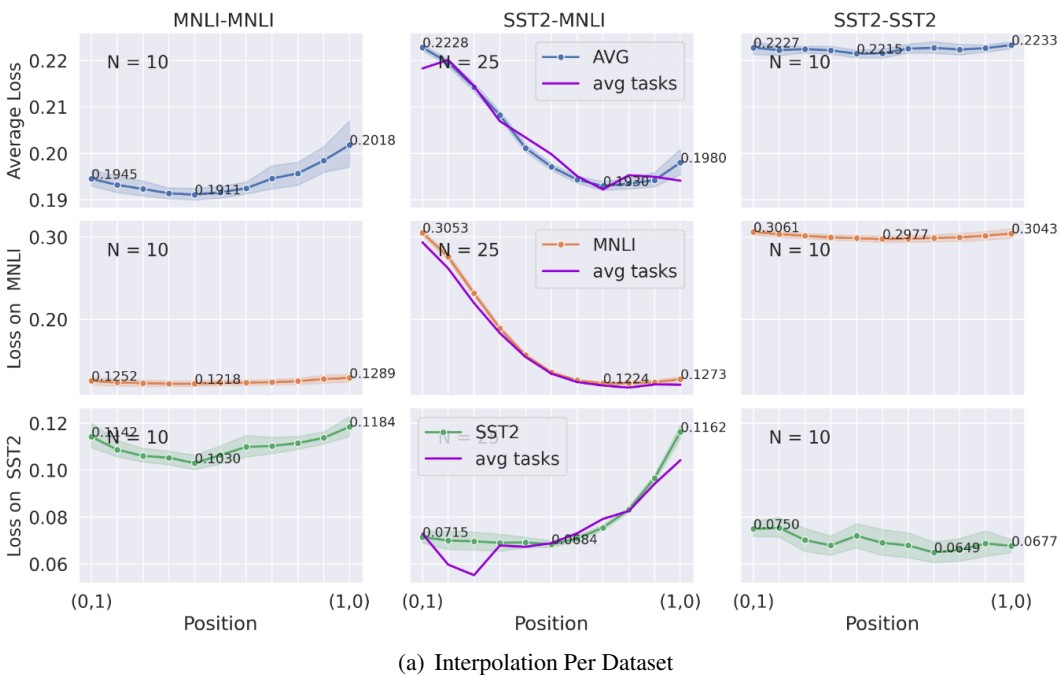

(a) Interpolation Per Dataset

Figure 10: Losses of linearly interpolated models created between pairs of similar models. In each figure, the solid line is the losses' average during interpolations for different $\alpha$ values, the edges of the lines represent the pure finetuned models we interpolated, Y axis is the average loss value, X axis is the position determined by $\alpha$, N is the number of pairs we interpolated between. The minimum average loss during the interpolation is noted and the shade is the standard deviation of the losses average. The purple line provides the average loss of the interpolation between centroids of models.

| dataset name | boolq | cb | cola | mnli | mrpc | multirc | qnli | qqp | rte | sst2 | wic | wnli | mean |
|---|---|---|---|---|---|---|---|---|---|---|---|---|---|
| Pretrain | 62.17 | 50.36 | 69.13 | 53.17 | 68.38 | 57.20 | 64.88 | 74.49 | 50.40 | 78.78 | 55.14 | 54.08 | 61.51 |
| Fuse | 62.23 | 56.79 | 69.49 | 63.01 | 69.46 | 57.14 | 73.77 | 79.91 | 61.59 | 84.91 | 55.52 | 52.68 | 65.54 |
| Gain | 0.06 | 6.43 | 0.36 | 9.85 | 1.08 | -0.06 | 8.89 | 5.42 | 11.19 | 6.12 | 0.38 | -1.41 | 4.03 |

Table 2: Gains of efficient finetuning starting from the centroid or the pretrained model. In columns names of datasets (mean is their average) and in rows the choice of base model and their difference, the gain.

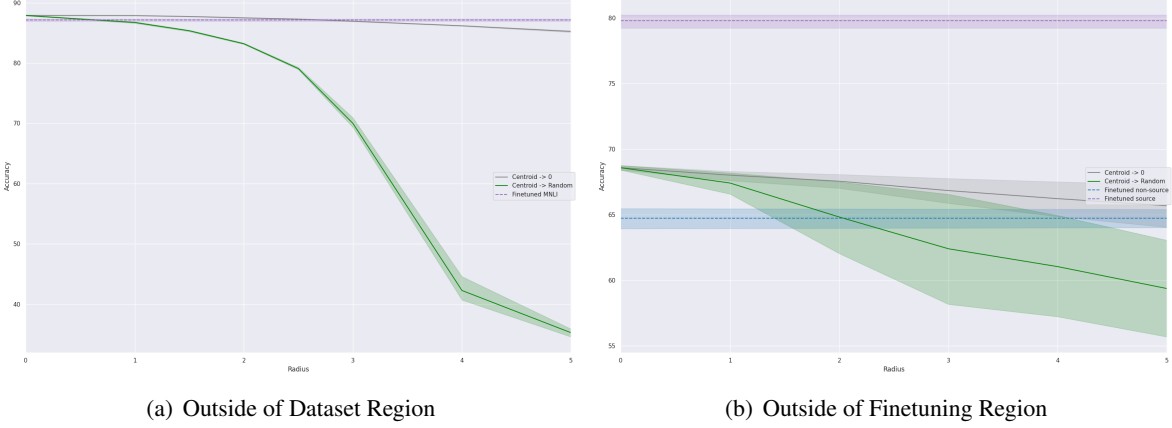

(a) Outside of Dataset Region          (b) Outside of Finetuning Region

Figure 11: Performance of the finetuned and the generated models from the centroid to the origin and to random directions, with respect to the distance from the region. In each graph, Y axis is the accuracy, X axis is the radius (which is the $\alpha$ values used for generating the models. Only relevant for the constant lines), the solid lines present the average accuracy of the generated models, the dashed lines present the average accuracy of the finetuned models (a constant value), and the shade is the standard deviation of the accuracies average. Models' groups in legend.

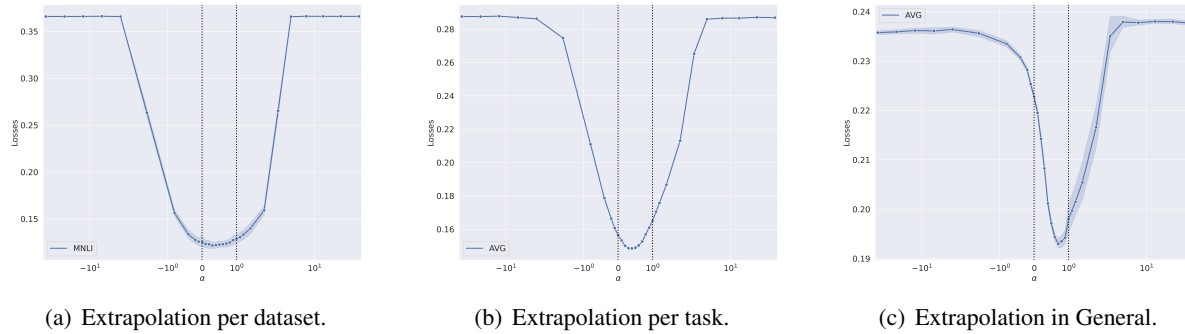

(a) Extrapolation per dataset.       (b) Extrapolation per task.       (c) Extrapolation in General.

Figure 12: Losses of linearly extrapolated models created from pairs of similar models. In each figure, the solid line is the average losses during extrapolations for different $\alpha$ values, the vertical dashed lines indicate the average loss of the pure models we extrapolate ($\alpha = 0$ or $\alpha = 1$), the Y axis is the average loss value, and the X axis is the position (meaning the $\alpha$ and $(1 - \alpha)$ values used in the extrapolation). The shade is the standard deviation of the losses' average across runs.

| dataset name | boolq | cb | cola | mnli | mrpc | multirc | qnli | qqp | rte | sst2 | wic | wnli | mean |
|---|---|---|---|---|---|---|---|---|---|---|---|---|---|
| Pretrain | 62.17 | 50.36 | 69.13 | 34.04 | 68.38 | 57.20 | 50.72 | 63.18 | 48.52 | 50.92 | 49.91 | 54.08 | 54.88 |
| Fuse | 62.23 | 56.79 | 69.49 | 63.01 | 69.46 | 57.14 | 73.77 | 79.91 | 61.59 | 84.91 | 55.52 | 52.68 | 65.54 |
| Gain | 0.06 | 6.43 | 0.36 | 28.97 | 1.08 | -0.06 | 23.04 | 16.74 | 13.07 | 33.99 | 5.61 | -1.41 | 10.66 |

Table 3: Gains of efficient finetuning with up to 1K examples, starting from the centroid or the pretrained model. In columns names of datasets (mean is their average) and in rows the choice of base model and their difference, the gain.

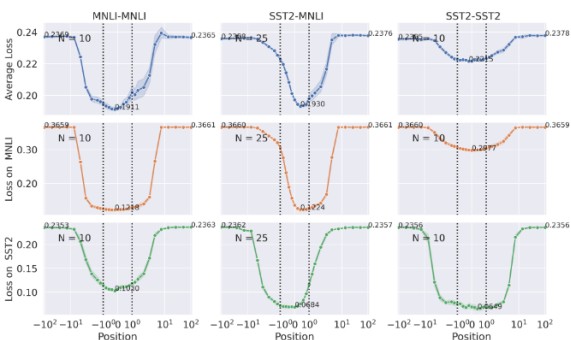

(a) Extrapolation Per Task and mixed

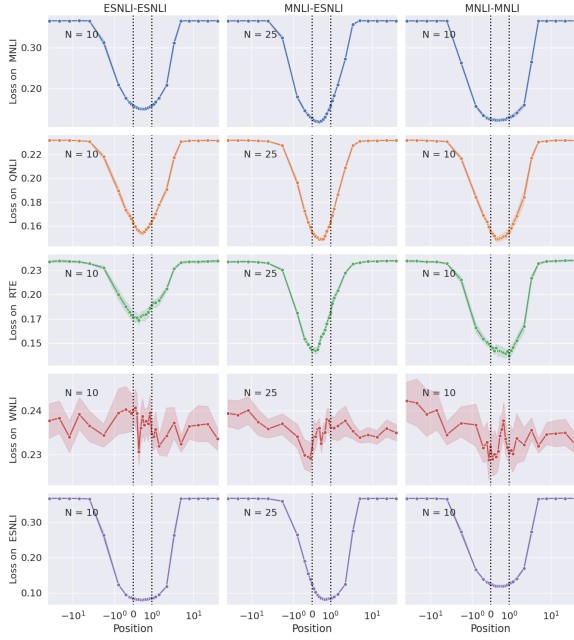

(b) Extrapolation Per Domain

Figure 13: Losses of linearly extrapolation models created between pairs of similar models. In each figure, the solid line is the average losses during extrapolations for different $\alpha$ values, the vertical dashed lines indicate the average loss of the pure models we extrapolate ($\alpha = 0$ or $\alpha = 1$), Y axis is the average loss value, X axis is the position (meaning the $\alpha$ and $(1 - \alpha)$ values used in the extrapolation), N is the number of pairs we extrapolated between, the values on top of the line are the loss at the edges and at the minimum average loss during the extrapolation, and the shade is the standard deviation of the losses average. Each Column represents extrapolation between different types of models and each row evaluates those same models and their extrapolations on a different target tasks.

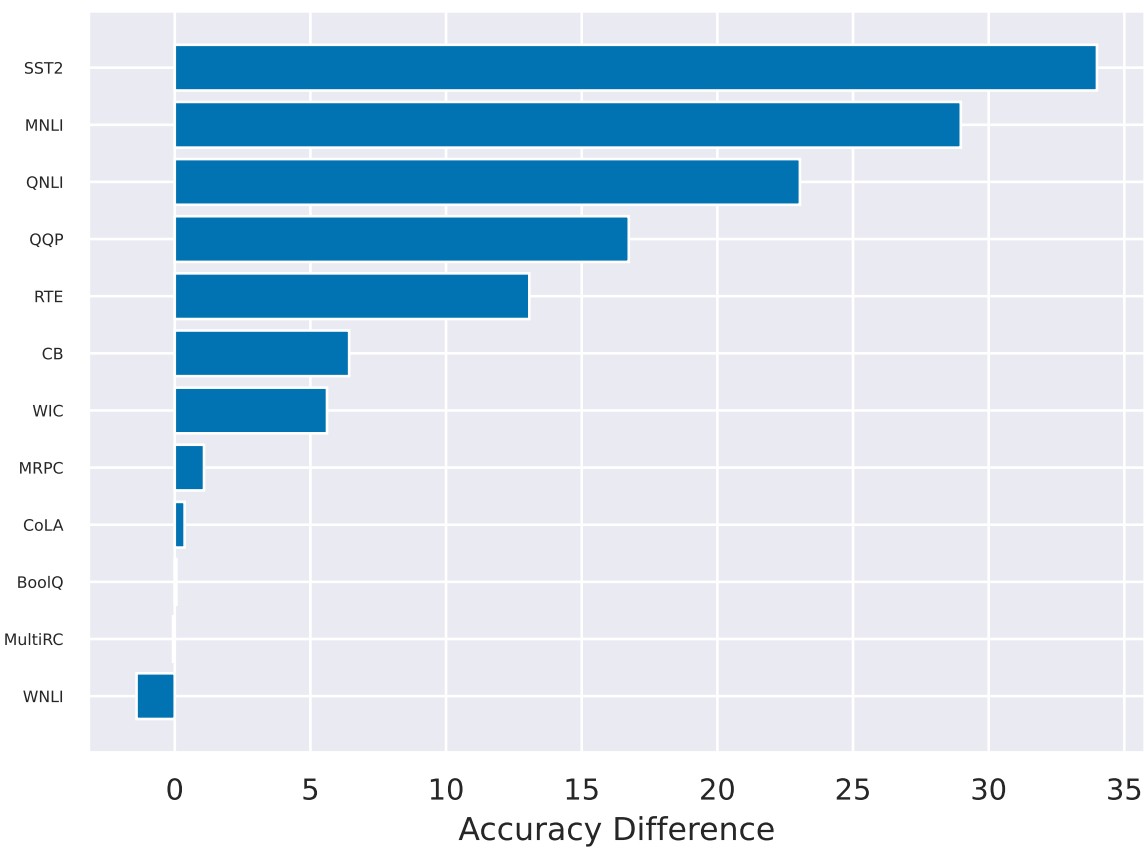

Figure 14: Losses of pretrained and centroid models on several target datasets, where both models were efficiently finetuned using BitFit in a few-shot scenario limiting training data to 1K.