# OpenReview forum: "Knowledge is a Region in Weight Space for Fine-tuned Language Models"
_EMNLP/2023/Conference — EMNLP 2023 Findings_

### Official Review · Reviewer_TEP2 · 2023-07-31

**Soundness:** 3

**Excitement:**

4: Strong: This paper deepens the understanding of some phenomenon or lowers the barriers to an existing research direction.

**Missing References:**

[1] Linear Connectivity Reveals Generalization Strategies. Juneja et al. ICLR23.
[2] Improving generalization of pre-trained language models via stochastic weight averaging. Peng et al. EMNLP22.
[3] Uncertainty-Aware Natural Language Inference with Stochastic Weight Averaging. Talman et al. NoDaLiDa23.

**Paper Topic And Main Contributions:**

This work explores the relationship between fine-tuned language models in weight space empirically. They found that models fine-tuned on the same dataset cluster together in the weight space and models fine-tuned on the same task also form clusters though looser ones. Models fine-tuned on any language tasks seem to reside in a constrained region compared to random perturbations of pre-trained weights.

**Questions For The Authors:**

1. This paper discussed a lot of the weight space of full parameters. Why do you compare your centroid initialization with pre-training initialization for 'efficient fine-tuning [1]'? What is the performance comparison of your fuse method and regular finetuning (full-parameter)?

2. What is the point if you have to fine-tune the full parameters of several language models first to get the centroid and then perform efficient fine-tuning [1]?

3. How do you handle the trainable parameters of layer-norm layers when you do interpolation/extrapolation of different models?

[1] BitFit: Simple parameter-efficient fine-tuning for transformer-based masked language models. Zaken et al  ACL22

**Reasons To Accept:**

1. The paper investigated an important problem: understanding relationships between fine-tuned models in weight spaces.
2. They conducted experiments on extensive datasets and analyzed the interpolation and extrapolation of language model encoders.
3. Their findings are well-motivated and thoroughly demonstrated through detailed experiments and visualizations. For example, the comparison between interpolation and extrapolation in the weight space significantly shows the properties of different regions.

**Reasons To Reject:**

1. The baseline results in Table 2 are pretty poor compared to both full-parameter and partial-parameter fine-tuning. For instance, MNLI: (53.17), QNLI: ( 64.88), QQP: (74.49), MRPC: (68.38), and SST2: (78.78). Although BitFit [1] use BERT models, the performance gap is quite large, In their paper, BitFit can achieve on-par performance as full-parameter fine-tuning. Could you explain it?

2. The motivation to use the centroid of fine-tuned full-parameters models as the initial point of efficient fine-tuning is missing.

Overall, I think this is a solid empirical analysis work studying the weight space of the fine-tuned language models, the authors provide enough analysis and show interesting findings to this community. It seems they want to propose a method based on their findings, however, I think this part is not well-motivated and the experiments are weak. Besides, several related works are missing.

**Reproducibility:**

3: Could reproduce the results with some difficulty. The settings of parameters are underspecified or subjectively determined; the training/evaluation data are not widely available.

**Reviewer Confidence:**

4: Quite sure. I tried to check the important points carefully. It's unlikely, though conceivable, that I missed something that should affect my ratings.

---

> ### Author Rebuttal · Authors · 2023-08-28
>
> We thank you for the review and deep reading, and for pointing out that the paper “shows interesting findings to this community”. In general, we are of one thought that the main part of the paper is the analysis, and the last practical part is mostly an example of how this might be beneficial, We also mention other cases where this explains previous practical findings.
>
> Regarding your questions:
> 1. Regarding the two questions about weight space of full parameters:
>     * Motivation for using the centroid (also asked about in reason to reject  #2) - Thank you, we will elaborate on it in the camera-ready version of the paper. Essentially, our findings suggest that certain regions perform well, and it is not clear what in them performs better (but not the edges or outside them). As there are endless possibilities of points in this region, we presumed the centroid is a bit less of an arbitrary choice than others, it might also avoid leaning towards any specific model and for us was the intuitive choice without adding further information. Therefore, for efficient fine-tuning we selected the centroid as a single, practical and efficient initialization point.
>     * Regarding the comparison to efficient\regular fine-tuning -
> When considering (efficient) fine-tuning, we see the choice of fine-tuning method and initialization as two complementary choices. The user chooses the fine-tuning method, and we provide a preferable initialization to fine-tune.
> In terms of comparing to a finetuned model -- even without fine-tuning, the fused models already outperform the finetuned models on the task. We show in sections 5.1 and 5.2 that the fused models (In’) perform comparably or even better than the regular fine-tuned ones (In) which they are created from, even though the In’ models are not fine-tuned by themselves.
> This implies that full fine-tuning would be even better, as previous papers indeed show. We cited them in sections 5.2 and 8, but we will emphasize this in the camera-ready version.
>
> 2. Regarding efficiently fine-tuning finetuned models- We fine-tuned the full parameters of several language models for the purposes of the research, to get a set of models that we know their training settings and procedure, and to avoid possible confounds. However, in a practical setting one can take existing fine-tuned models, for example from Hugging Face or users of the same service, and use them to define the region. Thus, the only computation done would be to average (CPU) and efficiently finetune.
>
> 3. Regarding the layer-norm parameters- In practice, we had no issues with them and therefore we did not need to handle them separately. We assume that their norm is similar enough, as they started from the same pre-trained model and did not diverge too much. Some papers deal with fusing norm-layers (cited in the related work), but it is always in the context of training from scratch where this issue is more critical.
>
> Regarding missing citations - thank you for noting that, we will add these papers in the camera-ready version.
>
> Regarding the BitFit results - it is an interesting point we did not note while writing the paper as we focused on comparing the two BitFit cases to each other.
>
>
> We are sorry it got a bit long. We have tried to address all your concerns in a way that respects the depth of your questions, and believe that our paper has improved considerably by emphasizing those points in our writing.

---

### Official Review · Reviewer_xwZQ · 2023-08-09

**Soundness:** 3

**Excitement:**

4: Strong: This paper deepens the understanding of some phenomenon or lowers the barriers to an existing research direction.

**Paper Topic And Main Contributions:**

This paper explores the relationship between the weight space and loss landscape of fine-tuned language models, and how to navigate the region between models to achieve better performance. The authors studied the weight space and loss landscape of fine-tuned models by training on 36 datasets and found similarities and differences between the fine-tuned models. Additionally, they proposed a new method to visualize the similarities between fine-tuned models and discussed the limitations of fine-tuning techniques in deep learning. The theme of this paper is fine-tuning techniques in deep learning, with a focus on the relationship between the weight space and loss landscape of fine-tuned models.

**Questions For The Authors:**

1. The experimental dataset of the paper is mainly concentrated on English classification data, can the similarities and differences between fine-tuned models be studied on datasets of other languages and fields?

**Reasons To Accept:**

1. This paper conducts in-depth research on the relationship between the weight space and loss landscape of fine-tuned language models, providing new ideas and methods for the study of deep learning.
2. By fine-tuning 36 datasets, the authors discover the similarities and differences between fine-tuned models and propose a new method to visualize the similarities between fine-tuned models.
3. The paper proposes a new method to evaluate the performance of fine-tuned models, which can more accurately assess the performance of fine-tuned models. Besides, this paper explores the limitations of fine-tuned models, providing new ideas and methods for the study of deep learning.

**Reasons To Reject:**

1. Although the experimental results of the paper show that the similarities and differences between fine-tuned models have an important impact on the performance of the models, further research is needed on how to use these findings to improve the performance of the models and explain these findings.


**Reproducibility:**

4: Could mostly reproduce the results, but there may be some variation because of sample variance or minor variations in their interpretation of the protocol or method.

**Reviewer Confidence:**

3: Pretty sure, but there's a chance I missed something. Although I have a good feel for this area in general, I did not carefully check the paper's details, e.g., the math, experimental design, or novelty.

---

> ### Author Rebuttal · Authors · 2023-08-28
>
> We thank you for the positive feedback and for finding many reasons for accepting the paper.
>
> Regarding your question about other languages and fields:
> We assume they behave in a similar way to the English classification datasets, as it is unlikely that the similarities and differences between fine-tuned models are language- or field-specific. However, in order to deeply research the relationship between the weight space and loss landscape, we chose to focus on a specific language and field and leave other datasets for future work. It is worth noting that the datasets we studied cover a wide range of tasks, and we observed the same similarities and differences in all of them. This also supports the belief that this will be the case for other datasets. While not strictly applying the same methodology, an ongoing work of ours tries to shed more light on multilingual model weights. We see that as a separate branch of work.
>
> Regarding your comment about further research on how to use these findings to improve the performance of the models and explain these findings, we definitely agree. We consider our work as an important first step in researching these findings and hope it will motivate future work on explaining them and finding more ways to exploit them. We would like to point out that we elaborated on several practical improvements and demonstrated one of them in Section 7.

---

### Official Review · Reviewer_JyVr · 2023-08-10

**Soundness:** 3

**Excitement:**

3: Ambivalent: It has merits (e.g., it reports state-of-the-art results, the idea is nice), but there are key weaknesses (e.g., it describes incremental work), and it can significantly benefit from another round of revision. However, I won't object to accepting it if my co-reviewers champion it.

**Paper Topic And Main Contributions:**

The authors study how the weight space and the underlying loss landscape of different models are interconnected. Based on their findings, the authors propose a better method for selecting a model for fine-tuning.

**Reasons To Accept:**

1) Very interesting perspective on weight space analysis, clustering/visual analysis look very explanatory.
2) Observation that well-performing models are centered in certain areas of the weight space might be useful for potentially selecting the final model (as opposed to standard evaluation).

**Reasons To Reject:**

1) The main message of the paper is still not very clear - how can we exploit this finding (well-performing models form a tight cluster in weight space)? Is this always true?
2) It would be good to provide a list of useful fine-tuning practices based on authors' findings.

**Reproducibility:**

4: Could mostly reproduce the results, but there may be some variation because of sample variance or minor variations in their interpretation of the protocol or method.

**Reviewer Confidence:**

2: Willing to defend my evaluation, but it is fairly likely that I missed some details, didn't understand some central points, or can't be sure about the novelty of the work.

---

> ### Author Rebuttal · Authors · 2023-08-28
>
> We thank you for the positive review and for finding the paper as interesting and useful.
>
> Regarding your questions:
> 1. The main message of the paper is as you stated at the beginning, but this is not a method paper. We do show that our findings have practical implications but do not presume to say this is the central contribution or the main point of the paper. We believe this is also the source of confusion.
>
>     * One way to exploit this finding (well-performing models form a tight cluster in weight space) is to select the center of the cluster as a starting model for fine-tuning, as you mentioned in the paper’s contributions. Additionally, since models inside the region share partial information with other models in the region (and outperform even the edges of the region), using any model from the region as a starting model should benefit the user. Another possible way to leverage our insights is to average models from the same region and expect the resulting model to perform better in the Same-Dataset region, without further fine-tuning.
>
>     * Guaranteeing that this finding will always be true is hard. While our experiments included thousands of fine-tuned models, trained on 36 datasets and also evaluated on 36 datasets, and we observed the same behaviour of tight clusters in all of them, we did not replicate it on many pretrained models yet. We believe future research could continue to investigate this behaviour and possibly find a promise or conditions for it.
>
> 2. We agree that it is beneficial to provide a list of fine-tuning practices and elaborate on practical implications in Section 7. We included several practices and demonstrated a possible practice for subsequent training and its results.

---

### Official Review · Reviewer_UZHb · 2023-08-12

**Soundness:** 4

**Excitement:**

4: Strong: This paper deepens the understanding of some phenomenon or lowers the barriers to an existing research direction.

**Missing References:**

N/A

**Paper Topic And Main Contributions:**

This paper investigates the relationships between different models, particularly those trained or tested on different datasets. While most previous research has concentrated on understanding individual models trained on specific datasets, this study delves into the connections between various models trained or tested on distinct datasets. The findings highlight that a model positioned between two similar models can assimilate the knowledge of both, enabling the authors to propose an efficient fine-tuning method that starts from the center of the cluster, outperforming the traditional approach in terms of accuracy.

**Questions For The Authors:**

See **Reasons To Reject**

**Reasons To Accept:**

The paper offers a unique and intriguing perspective by investigating the relationships between different models in the weight space and loss landscape. This sheds light on the transferability of knowledge between models and provides a basis for improving fine-tuning processes.

The authors provide solid empirical evidence to support their claims. Through experiments and analyses, they demonstrate the existence of clustered regions in the weight space, the improved performance of models within these clusters, and the potential for generating high-performing models through traversal within these regions.

The proposed fine-tuning method based on starting from the cluster center is a practical contribution. It offers a systematic approach to leveraging the knowledge contained within the weight space clusters.


**Reasons To Reject:**

While the empirical evidence is compelling, the paper could benefit from a deeper theoretical exploration of why these weight space clusters exist and how they facilitate knowledge transfer.


**Reproducibility:**

4: Could mostly reproduce the results, but there may be some variation because of sample variance or minor variations in their interpretation of the protocol or method.

**Reviewer Confidence:**

4: Quite sure. I tried to check the important points carefully. It's unlikely, though conceivable, that I missed something that should affect my ratings.

**Typos Grammar Style And Presentation Improvements:**

It is advisable to use consistent annotation and font style, e.g. line 415-417 $ln$ and $\text{ln}$

---

> ### Author Rebuttal · Authors · 2023-08-28
>
> We thank you for your support, as well as the clear and accurate summary that implies that our submission was well-understood.
>
> We agree regarding the benefit of a deeper theoretical exploration of why these weight space clusters exist and how they facilitate knowledge transfer. At this point, we can only demonstrate the existence of these regions in hopes of motivating future work towards understanding why these regions exist and how they correspond to knowledge transfer. We consider our work as an important first step in identifying this characteristic that will motivate future work on exploring it deeper.
>
> Thank you for your style improvements, we will fix those to ensure a consistent style.

---

### Meta-Review · Area_Chair_jcL8 · 2023-09-19

**Recommendation:** 4

**Metareview:**

This paper studies how the weight space and underlying loss landscape of different models are interconnected. The authors have provided insightful findings as summarized well by one of the reviewers that "They found that models fine-tuned on the same dataset cluster together in the weight space and models fine-tuned on the same task also form clusters though looser ones. Models fine-tuned on any language tasks seem to reside in a constrained region compared to random perturbations of pre-trained weights." Based on these analysis, the authors propose to create a better model by picking center of a region in the weight space, resulting in superior results. As summarized by the reviewers and myself, the pros and cons are:

### Pros:
1. This paper provides very novel and insightful empirical analysis as well as discussion on the weight space and loss landscape of different fine-tuned models. This is relatively new yet important direction for knowledge transfer.
2. "The authors provide solid empirical evidence to support their claims. Through experiments and analyses, they demonstrate the existence of clustered regions in the weight space, the improved performance of models within these clusters, and the potential for generating high-performing models through traversal within these regions."

### Cons:
1. "how to use these findings to improve the performance of the models and explain these findings."
2. Baseline results in Table 2 are poor.

Most of the reviewers agree that this paper is sound and exciting. Even though that this paper does not demonstrate the practical application of the findings in a comprehensive way, I think this analysis itself is interesting and insightful enough to stand as a nice paper alone.

---

### Decision · Program_Chairs · 2023-10-07

**Decision:**

Accept-Findings

**Comment:**

This paper studies how the weight space and underlying loss landscape of different models are interconnected. The authors have provided insightful findings as summarized well by one of the reviewers that "They found that models fine-tuned on the same dataset cluster together in the weight space and models fine-tuned on the same task also form clusters though looser ones. Models fine-tuned on any language tasks seem to reside in a constrained region compared to random perturbations of pre-trained weights." Based on these analysis, the authors propose to create a better model by picking center of a region in the weight space, resulting in superior results. As summarized by the reviewers and myself, the pros and cons are:

### Pros:
1. This paper provides very novel and insightful empirical analysis as well as discussion on the weight space and loss landscape of different fine-tuned models. This is relatively new yet important direction for knowledge transfer.
2. "The authors provide solid empirical evidence to support their claims. Through experiments and analyses, they demonstrate the existence of clustered regions in the weight space, the improved performance of models within these clusters, and the potential for generating high-performing models through traversal within these regions."

### Cons:
1. "how to use these findings to improve the performance of the models and explain these findings."
2. Baseline results in Table 2 are poor.

Most of the reviewers agree that this paper is sound and exciting. Even though that this paper does not demonstrate the practical application of the findings in a comprehensive way, I think this analysis itself is interesting and insightful enough to stand as a nice paper alone.